# The Diet Quality and Nutrition Inadequacy of Pre-Frail Older Adults in New Zealand

**DOI:** 10.3390/nu13072384

**Published:** 2021-07-13

**Authors:** Esther Tay, Daniel Barnett, Evelingi Leilua, Ngaire Kerse, Maisie Rowland, Anna Rolleston, Debra L. Waters, Richard Edlin, Martin Connolly, Leigh Hale, Avinesh Pillai, Ruth Teh

**Affiliations:** 1Department of General Practice and Primary Health Care, School of Population Health, University of Auckland, Auckland 1023, New Zealand; esther.tay@auckland.ac.nz (E.T.); e.leilua@auckland.ac.nz (E.L.); n.kerse@auckland.ac.nz (N.K.); 2Department of Statistics, Faculty of Science, University of Auckland, Auckland 1010, New Zealand; Daniel.Barnett@auckland.ac.nz (D.B.); a.pillai@auckland.ac.nz (A.P.); 3Human Nutrition Research Centre, Population Health Sciences Institute, Faculty of Medical Sciences, Newcastle University, Newcastle Upon Tyne NE2 4HH, UK; Maisie.Rowland@newcastle.ac.uk; 4The Centre of Health, Tauranga 3110, New Zealand; a.rolleston@auckland.ac.nz; 5Department of Medicine, School of Physiotherapy, University of Otago, Dunedin 9016, New Zealand; debra.waters@otago.ac.nz; 6Health Systems Group, School of Population Health, University of Auckland, Auckland 1023, New Zealand; r.edlin@auckland.ac.nz; 7Department of Geriatric Medicine, Faculty of Medical and Health Sciences, University of Auckland, Auckland 1023, New Zealand; Martin.Connolly@waitematadhb.govt.nz; 8Waitematā Clinical Campus, Waitematā District Health Board, Auckland 0622, New Zealand; 9Centre for Health, Activity and Rehabilitation Research, School of Physiotherapy, University of Otago, Dunedin 9016, New Zealand; leigh.hale@otago.ac.nz

**Keywords:** frailty, macronutrients, micronutrients, aged, diet quality

## Abstract

This study aimed to describe the diet quality of pre-frail community-dwelling older adults to extend the evidence of nutrition in frailty prevention. Pre-frailty, the transition state between a robust state and frailty, was ascertained using the FRAIL scale. Socio-demographic, health status, and 24-h dietary recalls were collected from 465 community-dwelling adults aged 75+ (60 years for Māori and Pacific people) across New Zealand. Diet quality was ascertained with the Diet Quality Index-International (DQI-I). Participants (median (IQR) age 80 (77–84), 59% female) had a moderately healthful diet, DQI-I score: 60.3 (54.0–64.7). Women scored slightly higher than men (*p* = 0.042). DQI-I components identified better dietary variety in men (*p* = 0.044), and dietary moderation in women (*p* = 0.002); both sexes performed equally well in dietary adequacy and poorly in dietary balance scores (73% and 47% of maximum scores, respectively). Low energy 20.3 (15.4–25.3) kcal/kg body weight (BW) and protein intakes 0.8 (0.6–1.0) g/kg BW were coupled with a high prevalence of mineral inadequacies: calcium (86%), magnesium (68%), selenium (79%), and zinc (men 82%). In conclusion, the diet quality of pre-frail older adults was moderately high in variety and adequacy but poor in moderation and balance. Our findings support targeted dietary interventions to ameliorate frailty.

## 1. Introduction

The prevalence of frailty increases with age. It is suggested that the prevalence of pre-frailty and frailty in community-dwelling older adults (those aged 65 and older) is about 42% and 11% [1]. Those in a pre-frail state, where this is defined as a transitional state between robust and frailty [2,3], may be targets for intervention to potentially reverse age-related disability. Within the complex and dynamic “cycle of frailty”, chronic undernutrition, sarcopenia and obesity, and energy and nutrient imbalance show that diet quality plays an important role in influencing frailty prognosis [3].

Studies have reported on energy, protein (distribution, quality, source, leucine), and micronutrients with antioxidant/anti-inflammatory activity (vitamin D, E, C, folate, PUFA n-3) being the major findings associated with frailty [4]. Single food or nutrient research is limited as it does not consider the lived reality of eating which involves meals and patterns. This single food or nutrient approach also misses the complex interactions between nutrients and food constituents which can have significant impact on health outcomes [5,6]. To add to the evidence base of nutrient studies, a growing number of frailty-oriented studies have used holistic diet quality indices to assess the diet quality in the frail population group.

A comprehensive systematic review on the dietary patterns of frail older people was reported by Fard et al. in 2019, which demonstrated that in the cross-sectional and prospective studies, most indices have shown inverse relationships between diet quality and frailty [7]: The most common diet quality index that showed this negative relationship was the Mediterranean diet score (MDS [8,9,10], aMED [11,12,13]). Other indices showing similar results were: the Healthy Eating Index (HEI) [14], and alternates (aHEI [10], aHEI-2010 [13]), Dietary Approaches to Stop Hypertension (DASH) [10,13], Elderly Dietary Index (EDI) [15], Dietary Inflammatory Index (DII) [16], Diet Quality Index revised and International (DQI-R [17], and DQI-I [18], respectively). Some studies however, found no associations: the Healthy Diet Indicator (HDI) [15] and MDS [18]. The variations in the findings are likely to be attributed to assessment methods and dietary culture [7].

Hlaing-Hlaing et al. conducted a systematic review in 2017 to understand the breadth of diet quality indices used in adults in Australia and New Zealand [19]. Researchers found that of the 25 indices identified, only one was applied in New Zealand; the Healthy Dietary Habits Index (HDHI), and this was for adolescents [20]. In the reviewer’s critical appraisal, most of the indices used in Australia and New Zealand had been built for detecting over-nutrition [19]. Trijsburg et al., in their systematic review, suggested indices to cover both over- and under-nutrition to be applicable for different population groups and global comparison [21].

The DQI-I has many advantages over other indices currently used in the literature in that it accounts for both under- and over-nutrition and incorporates national and international dietary guidelines and thus enables specific population assessment and cross-country comparisons [22,23]. In addition, the DQI-I accounts for dietary variety which many other indices do not have [23] and utilises a nested structure so that target problem areas of diet quality can be identified alongside the total score measure. The index was also constructed for 24-h recall rather than food frequency questionnaire data which is of benefit as the latter has been found to underestimate unhealthy eating in older people [24]. Lastly, the index has previously been effective in determining frailty risk; Chan et al. demonstrated that an increase of 10 points in DQI-I score was correlated with a 41–59% reduced risk of incident frailty over four years follow-up in older adults (*n* = 2724, mean age 71.8, SD 4.8) living in Hong Kong [18].

The evolving evidence of nutrition in frailty pathogenesis stresses the importance of assessing the nutritional status of pre-frail older adults [2,4,25]. To date, few studies have focused on determining the diet quality in the pre-frail population group, and to the best of our knowledge, none have been conducted on the pre-frail population in New Zealand. The Staying UPright and Eating well Research (SUPER) study in New Zealand was conducted in 2016 to 2020 to test the effectiveness of nutrition, exercise, or social groups in ameliorating the progression to frailty among a population of pre-frail community-dwelling older adults [26]. This paper aimed to examine the diet quality and nutrition adequacy of this population group to inform nutrition interventions for frailty prevention.

## 2. Materials and Methods

### 2.1. Study Sample

The SUPER study recruited 468 pre-frail older adults living in the community across New Zealand through general practices. Older adults were defined as those aged 75 and over (60 years for Māori and Pacific people). A lower age group was chosen for Māori and Pacific people as there is disparity in health status and life expectancy compared to the general population and levels of disability occur at a younger age for these groups [27]. Pre-frailty was defined as a score of 1 to 2 in the FRAIL scale screening tool further described in detail elsewhere [26]. The study was approved by the Southern Health and Disability Ethics Committee, Ministry of Health, New Zealand (Ref 14/STH/101/, 13 August 2014).

### 2.2. Data Collection

The SUPER study measured dietary intake using a 24-h recall based on the multiple pass method, for two non-consecutive days in a sample of 468 pre-frail older adults living in the community. The multiple pass method records detailed descriptions of dietary intake and is found to be suitable for the general [28] and the oldest old population [29].

Trained interviewers conducted a face-to-face interview using an online 24-h recall tool, Intake24, to complete food records and demographic information. Intake24 was developed by Newcastle University (UK) and uses food photographs to estimate portion sizes to enhance the accuracy of dietary recalls [30]. Intake24 was adapted for New Zealand and the foods in the database linked to the New Zealand FOODfiles 2016 and Nutrient Databank from Public Health England databases [26,30].

The DQI-I assessment was used to determine diet quality based on its many advantages described above. The index incorporates four components—variety, adequacy, moderation, and overall balance—that add up to a maximum score of 100 points. Higher scores indicate better diet quality. The construction was described in detail by Kim et al. [22]. Adaptations to the DQI-I were used to cater towards the New Zealand population. For instance, instead of six unique protein sources, we used seven (including nuts and seeds) as the latest New Zealand Ministry of Health guidelines recognise nuts and seeds as part of the protein sources food group, which was not apparent in the original DQI-I [31]. The non-essential, energy-dense, nutritionally deficient foods in New Zealand (NEEDNT) food list was utilised and adapted for older people when determining empty calorie intake e.g., whole milk and yoghurt were removed from the list [32]. Assessment criteria and New Zealand-relevant adaptations are listed in Appendix A. Energy and nutrient descriptive values were computed, compared against New Zealand guidelines [31,33] and listed in Appendix A. Both tables were analysed by sex.

Socio-demographic, medical history and lifestyle behaviour were collected using standardised questionnaire by research-trained interviewers at participants’ residences: age, sex, marital status, education level, living arrangement, deprivation, medical conditions, vision, hearing, body mass index (BMI), medications, supplements, alcohol consumption and activities of daily living (the Nottingham Extended Activities of Daily Living, NEADL). Body composition data were measured using a Tanita BC-545N scale and Stadiometer: BMI was calculated as weight (kg)/height (m)^2^. The NEADL scale was used as a subjective measure to assess functional status. Deprivation was measured using participant addresses and standardised against the NZDep2018 [34]. More information on these measures is available in the study protocol [26].

### 2.3. Statistical Tests

#### 2.3.1. Descriptive Tests

Non-normally distributed data are presented as median (interquartile range (IQR)). Normally distributed data are presented as mean (standard deviation (SD)). Independent samples T-test was used to calculate normally distributed data (2-tailed) with 95% confidence intervals (CI). The non-parametric independent samples test was used to calculate non-normally distributed data, and Chi-square test for categorical variables.

#### 2.3.2. Data Sensitivity Analysis

Data sensitivity analysis was computed to account for dietary misreporting. Misreporting was identified using the Goldberg cut-off of 0.92 for under-reporters and 2.62 for over-reporters [35]. The Oxford equations were used to calculate basal metabolic rate (BMR) [36]. Calculations are detailed in Table 1. Univariate and multivariate analysis (multiple forward linear regression) were used to examine demographic, health variables (age, sex, marital status, education level, living arrangement, deprivation, medical conditions, vision, hearing, BMI, medications, supplements, alcohol consumption, and NEADL) and dietary variables associated with misreporting.

Statistical significance was set at *p* < 0.05. The Bonferroni correction was used to adjust for multiple testing [37]. All data from the baseline time-point were analysed using IBM SPSS Statistics for Windows, version 27 (IBM Corp., Armonk, NY, USA).

## 3. Results

In this sample of 468 participants, three did not complete the dietary assessments due to technical difficulties and two were missing weight and height data. The median age was 80 years (IQR: 77–84) and 58.9% of the sample was female.

### 3.1. Diet Quality

#### 3.1.1. Overall Findings

The diet quality of our pre-frail population group was around 60% of the maximum score. Females had a marginally higher median DQI-I score than males (females: 60.83 (IQR: 54.33–64.64); males: 59.27 (53.15–67.87), *p* = 0.042) (Figure 1, Appendix A).

As for DQI-I components, variety favoured males (68% versus 65% of maximum score, *p* = 0.044) and moderation favoured females (40% of maximum score for both sexes but males had a wider distribution of values, including some particularly high and low values when compared to the female group, *p* = 0.002). Both sexes performed equally well in adequacy (73% of maximum score, *p* = 0.142) and poorly in overall balance scores (47% of maximum score, *p* = 0.261).

As seen in Figure 1, participants scored the best in relation to adequacy, with this category contributing the biggest portion to the DQI-I total (40/100), followed by variety, balance and moderation components.

#### 3.1.2. Subcomponent Findings

Participants scored around 3.5 of 5 unique food groups a day according to the variety subcomponent, overall food group (Appendix A). As seen in Figure 2 below, while over 50% of participants met their daily recommended serving for major protein foods, up to 25% of them managed to meet their recommended servings of dairy, grains, fruit, and vegetables [31]. Men consumed more protein and grain food groups than women (Figure 2, *p* = 0.004 and *p* < 0.001 respectively). Sex differences were not seen for the remaining food groups.

Within-group variety subcomponent scores showed that two of three-plus unique sources of protein a day were consumed (Appendix A). The common sources of protein depicted in Appendix A demonstrate similarities between males and females. Over 77% obtained their protein from dairy sources, 28–45% from eggs and red meats, 15–23% from poultry, fish, and nuts/seeds and 6% from legumes.

For adequacy subcomponents, males scored better in iron (*p* < 0.001) compared to females, however both sexes met their recommended intake according to New Zealand guidelines (Appendix A, [31]). No other subcomponents differed by sex.

When looking into moderation and balance subcomponents, saturated fat and empty calorie intake were the major problem areas in our sample group’s diet scoring medians of 0 for both sexes (Appendix A). Saturated fat contributed to 13% of the total energy intake (TEI) i.e., above the upper limit guidelines for saturated fat and trans-fat combined of 10% TEI (Appendix A, [31]). Empty calorie foods contributed 36% (25–46) of the TEI compared to the DQI-I upper threshold of 10% (Appendix A, [22]).

Appendix A shows the contribution of macronutrients to TEI. Total energy intake from fat was close to the upper bounds of the acceptable macronutrient distribution range (AMDR) at 34% of the recommended 20–35% (Appendix A, [31]). The contribution of carbohydrate and protein were close to the lower boundary of the AMDR (47% of 45–65% and 16% of 15–25%, respectively). The poly-unsaturated to saturated fatty acid ratio (PUFA/SFA) was 0.22 and mono-unsaturated to saturated fatty acid ratio (MUFA/SFA) was 0.87 (data not shown).

### 3.2. Nutrition Inadequacy

When determining adequacy, the index focuses mainly on meeting requirements. As a secondary perspective, we looked at inadequacy, the other side of the coin. Males, unsurprisingly, had a 330 kcal higher energy intake than females (*p* < 0.001, Appendix A). However, when adjusting for body weight (BW), the energy intake was not significantly different (21.1 kcal/kg BW versus 19.7 kcal/kg BW). Protein intake for males and females was 0.81 g/kg BW and 0.78 g/kg BW, respectively (Appendix A).

In addition to low energy and protein intake, roughly 80% of males did not meet their estimated average requirements (EAR) for calcium, magnesium, selenium, and zinc (Appendix A). Findings were similar for females, with roughly 80% having inadequacy for calcium and selenium, 60% for magnesium and 45% for zinc. Compared to men, females had higher proportions of inadequacy for folate and thiamin, whilst men had higher inadequacy for vitamin A. The median vitamin D intake ranged around 2.5 μg a day, which is below the adequate intake (AI) of 15 μg a day (Appendix A). Similarly, vitamin E intake sat around 7.23 mg for males and 6.39 mg a day for females compared to the respective AI of 10 mg and 7 mg. The total sugar intake per day for males and females was 85.18 g (60.64–119.06) and 78.03 g (55.20–106.07), respectively.

### 3.3. Sensitivity Analysis: Misreporting

#### 3.3.1. Characteristics of Low Energy Reporters

One third (32.5%) of our participants were defined as low energy reporters (LER) [38,39] (Table 1). No over reporters were found in this sample.

Univariate analyses showed that LER were more likely to be of higher BMI than plausible reporters (31 kg/m^2^ versus 27 kg/m^2^) (*p* < 0.001) (Table 2). No differences were seen in age, sex, marital status, education level, housing, deprivation, vision, hearing, medications, supplements, alcohol consumption and NEADL score. As for diet quality variables, LER also scored lower in total DQI-I, variety, and adequacy and better in moderation compared to plausible reporters. No differences were seen between the two reporters for overall balance. Energy and protein per kg BW were lower in LER. Lastly, LER reported 7% lower energy contributions from empty calorie foods, fat, and saturated fat and a 2% higher contribution from protein than plausible reporters.

Multivariate analyses conducted against all listed demographic and health variables demonstrated that LER were more likely to have higher BMI (B = 0.034 (CI: 0.026–0.042), *p* < 0.001) and lower DQI-I total score (B = −0.007 (CI: −0.012–−0.002), *p* = 0.004) than plausible reporters (Appendix A)).

#### 3.3.2. Considerations of Low Energy Reporting

When excluding LER from DQI-I analyses, the significantly higher DQI-I and moderation scores of females was lost (male DQI-I median 60.2 (53.9–65.3); female 61.0 (56.05–65.2) (Table 3), *p* = 0.493 and male moderation median 35% (25–45) of maximum score; female 40% (30–45), *p* = 0.086). Males continued to have higher variety scores than females (male 75% (60–85) of maximum score; female 68% (60–79), *p* = 0.017). Adequacy and overall balance scores remained equal between sexes (male adequacy 76% (68–83) of maximum score; female 77% (70–82), *p* = 0.598 and male balance 45% (34–57); female 46% (33–55), *p* = 0.782).

Further details on the differences for subcomponents and nutrition inadequacies after LER exclusion are reported in the Appendix B.

## 4. Discussion

### 4.1. Diet Quality of Pre-Frail Older Adults

We aimed to examine the diet quality and nutrition adequacy in pre-frail community-dwelling older adults in New Zealand. According to the Diet Quality Index-International (DQI-I) and the threshold set by Kim et al. [22], our sample of pre-frail older adults had borderline poor quality diets. Until further studies are reporting on DQI-I, we cannot make adequate comparisons against other population groups. Higher scores in variety and adequacy components were balanced by poorer scores in moderation and balance, a pattern seen for both males and females (Appendix A).

Variety and adequacy subcomponents revealed that participants met many of their nutrient requirements such as fibre, protein, iron, and vitamin C. However, scores were poor with attainment to recommended servings of dairy and plant-based food groups: fruit, vegetables, and grains. 75% or less of participants did not meet serving recommendations for these food groups. Most participants met their single serving of protein a day by obtaining their protein from animal sources such as dairy, eggs, and red meat. Fewer participants obtained protein from lean meats or plant-based foods such as poultry, fish, nuts/seeds, and legumes. These protein sources are within the context of the DQI-I; in the 2008/9 New Zealand nutrition survey, bread was the main source of protein for older people [28].

The remaining 40% of the DQI-I score was accounted for by the moderation and balance components. Corresponding subcomponents revealed that discretionary food, particularly foods high in saturated fat, are notable features of the diet in pre-frail older adults. Empty calorie foods contributed three times over the upper limit of the threshold of 10% TEI set by the DQI-I.

Macronutrient distribution correspondingly was skewed towards higher fat and lower protein intakes. The overall 5% deviation from AMDR would be of interest to nutrition interventions to reduce fat and increase protein contributions. Non-red meat dependent protein sources as listed previously would provide a greater-balanced fatty acid profile from the current PUFA/SFA ratio of 0.22 and MUFA/SFA ratio of 0.87. As reflected in the DQI-I scoring criteria and existing literature, the ideal ratio of SFA:MUFA:PUFA for heart health appears to be 1:1.3:1 [40] deeming our population’s dietary fatty acid profile in need of improvement.

Although participants appeared to have adequate protein from the DQI-I score, protein intakes were below recommendations when adjusting for body weight (guidelines: males 1.07 g/kg BW; females 0.94 g/kg BW [31]). Protein intakes below 1 g/kg BW have been associated with a higher risk of frailty [41].

Total energy intake by body weight in our sample was 21.1 kcal/kg BW in males and 19.7 kcal/kg BW in females. This is below estimated requirements (25–30 kcal/kg BW) [42]. In addition, a previous study has shown that energy intake below the 21 kcal/kg BW threshold was associated with frailty [43]. A diet of low energy is likely to see inadequacies of macro- and micronutrients. Further analysis of nutrients showed inadequacies in calcium, magnesium and, selenium, alongside zinc for males and folate for females, according to New Zealand guidelines. Vitamin D and E also indicated deficiency, though firm conclusions cannot be made due to the absence of EAR against which to compare. These inadequacies are likely a direct result of the limited representation of plant-based foods in the diets reported. Given our participants are pre-frail, these findings do not come as a surprise as they reflect results from a notable cross-sectional study that suggested diets low in energy, protein and micronutrients were positively associated with risk of frailty [43]. However, as Hengeveld et al. found in their prospective study, energy, protein, and animal protein intake were not associated with frailty incidence over four years. Further to their findings, robust participants with lower plant-based protein intake had a higher risk of developing pre-frailty or frailty [14] suggesting that plant-based protein may be is as potent as animal protein in maintaining physical function. There are limited studies to date reporting the optimal level and physiological benefits of the animal and plant-based protein in delaying the exacerbation of frailty progression.

We also found a high consumption of empty calorie foods in this sample. Together with results on micronutrient inadequacies and excess saturated fat and sugar intake, these findings suggest the high consumption of empty calorie foods displaced consumption of micronutrient-rich foods such as fruits, vegetables, grain, dairy, and legumes. Plant-based foods such as beans, grains, fruits, and vegetables are good sources of the micronutrients that were below serving recommendations in this population. Incorporating an extra serving of fruits and vegetables and one to three extra servings of dairy, beans and grains would likely help meet micronutrient recommendations.

Lean meats such as poultry and fish as well as plant-based protein sources offer a wider variety of nutrients such as polyunsaturated fatty acids with the benefit of lower saturated fat. Increasing variety in this direction will help with the fatty acid profile which is in line with recommendations to protect cardiovascular health in older adults [31]. This is being said with consideration that dairy and red meats are still helpful to include in the diet for their provision of calcium and iron, which are important for older adults [31].

These findings are in line with current literature that suggests following a diet in closer adherence to the Mediterranean diet reduces the risk of frailty [8,9,10,11,12,13]. As the Mediterranean diet is characterised by a generous representation of plant-based foods alongside low-saturated fats, such as olive oil [44], pre-frail older adults can likely benefit from aligning their dietary pattern in this direction.

### 4.2. Sensitivity Analyses

As was the case in previously conducted nutrition surveys, one in three older adults were found to be LER [38,39]. LER were more likely to be of higher BMI and lower diet quality than plausible reporters independent of other demographic and health variables. LER in our sample were more likely to omit energy-dense and nutrient-scarce (empty calorie) foods from their dietary report which consequently produced a pseudo-higher score of moderation but compromised on variety, adequacy, and ultimately total DQI-I scores.

Sex differences were not seen after sensitivity analyses except for males scoring better in variety, which is likely due to their higher consumption of major protein and grain food groups compared to females.

Our main findings remain the same independent of LER, in that empty calorie foods, and unfavourable macronutrient and fatty acid distributions coupled with low attainment of dairy, grains, fruit, and vegetable serving recommendations were characteristic of this sample of pre-frail older adults. Our results found that the %TEI from protein was inflated in the LER, which was likely to be attributed to the lower reporting of energy intake, similarly reported in a systematic review [39].

Although deficiencies in micronutrients alongside energy and protein intake by body weight improved slightly upon LER exclusion, most remained deficient and below the recommendations. These findings indicate that firstly, the true dietary imbalance of nutrient-rich and empty calorie foods could be worse than represented in our study. Secondly, we found that absolute food intake is inadequate for our current understanding of requirements in this population, with further studies needed to assess appropriate recommendations.

Findings from DQI-I scores and nutrient analysis upon sensitivity analysis suggest that there is room for greater a variety of nutrient-rich foods as opposed to empty calorie foods alongside energy and protein for the dietary pattern of pre-frail older adults.

### 4.3. Strengths and Limitations

This paper is the first that we know of to assess the dietary intake of community-dwelling pre-frail older adults using the DQI-I coupled with nutrient analysis in New Zealand. The index is sensitive enough to identify undernutrition and overnutrition, which is seen in both developed countries such as New Zealand and developing countries [22]. In this regard, our results allow for future international comparisons. We still advise caution in generalising the findings to other populations which may have different dietary cultures and environments. Most of our study sample are of British descent and still follow the traditional homemade “Western diet” closely. As seen in the latest New Zealand nutrition survey (2008/9), less than 0.5% of adults aged over 71+ have fast food or takeaways three or more times a week [31]. It is therefore more likely that the dietary imbalance in New Zealand arises from a high consumption of ‘home foods’ such as biscuits, cakes, desserts, sweet drinks, sweets, butter and margarine, sweet spreads, meat or cheese pies and pastries, sauces, processed meats and alcoholic drinks [31,32].

Since low energy reporting is common in the oldest age group (33% of those over 65 years old) [38], our analyses also accounted for the likelihood of under- and over-reporting of dietary intake to minimise reporting bias in our conclusions.

As seen in this study and previously identified, the moderation and balance components of the DQI-I identified poor dietary qualities related to chronic and non-communicable disease [45]. Where the limitation of the DQI-I lie is in capturing micronutrient deficiencies, especially those relevant to disease-specific states, as seen in the paediatric allergen elimination diets [45] and, in our case, frailty prevention in older people. To manage this limitation, we looked at a broader range of micronutrients and measures of energy intake. Future studies should perhaps investigate adapting the DQI-I for capturing attainment of frailty-preventative dietary goals.

The current paper is unable to provide causal associations of diet quality with demographic and health indicators given the cross-sectional nature of the analysis. As we focus on diet quality, supplements were not considered in detail for the main analyses. The 44% of participants who took at least one supplement a day may have improved micronutrient intake. However, to accurately quantify nutrient intake from a wide range of supplements is beyond the scope of this paper. We also did not measure sun exposure, which accounts for some proportion of vitamin D provision.

The literature is growing but is inconclusive in pinpointing specific nutrients of concern for pre-frail older adults. Hence, conclusions are tentative. This study adds to the understanding of the nutritional gaps of those in this continuum so that guidelines and recommendations can be challenged and further strengthened in future research.

Perhaps the DQI-I algorithm needs to be adapted to ensure relevancy of diet quality in the ageing population. For instance, variety, adequacy, and moderation being highly correlated may cause specific areas of the diet to be weighted heavier than others [23]. The use of cholesterol does not provide as helpful information as other subcomponents, given that nutrition evidence shows serum cholesterol level (risk factor for cardiovascular disease) is not influenced by cholesterol intake [46]. Being the first to use this index in New Zealand, we take caution to interpret the scores due to potential population-specific differences. We recommend future studies to look at the potential of revising the DQI-I specifically for older adults and frailty prevention.

## 5. Conclusions

Our sample of pre-frail older adults showed diets that were moderately high in variety and adequacy but equivalently poor in moderation and balance. Micronutrients of target interest were calcium, magnesium, selenium, vitamin D, and Vitamin E alongside zinc for males.

The diet in our population was characterised by low energy and protein intake by body weight. Key areas of knowledge gaps remain specifically in the recommended threshold of energy, protein for pre-frail older adults.

Future studies may look to develop and test nutrition interventions that focus on rebalancing the diet with increased plant-based foods over empty calorie foods in the context of frailty. Such studies will help clinicians, policy makers and guidelines to be better equipped with causal evidence to ameliorate the effects of frailty in our ageing population.

## Figures and Tables

**Figure 1 nutrients-13-02384-f001:**
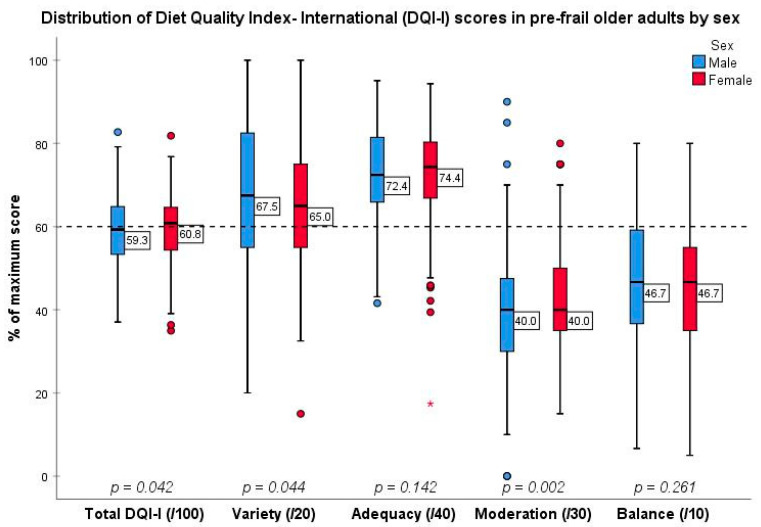
Box and whisker plot depicting DQI-I score attainment according to sex. *p*-values derived from non-parametric samples tests. The *p*-value for Balance score was derived by independent samples *t*-test. Each small circle represents a mild outlier to the data. The asterisk (*) represents an extreme outlier where a participant scored 17% of the maximum attainable score in adequacy.

**Figure 2 nutrients-13-02384-f002:**
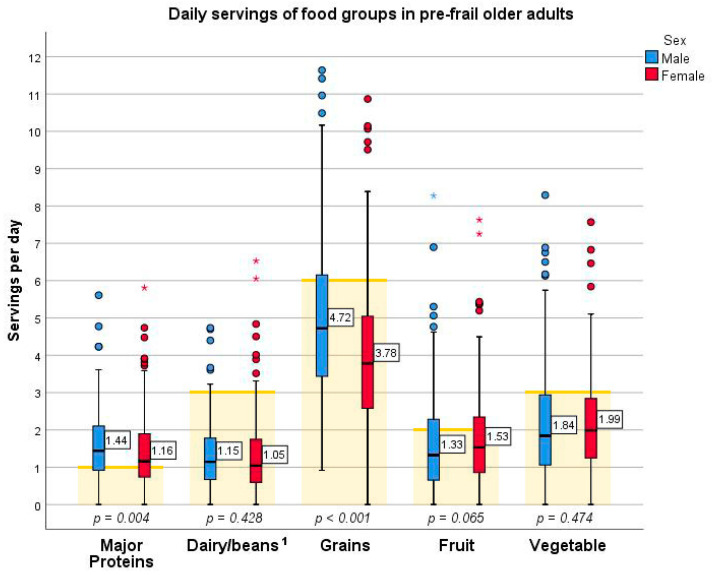
Box and whisker plot depicting daily servings of food groups consumed by pre-frail older adults by sex and overlaid on corresponding serving recommendations. Note: *p*-values represent differences between males and females and were determined by independent samples tests. Yellow bars represent recommended servings from New Zealand guidelines [31]. ^1^ Beans have been grouped with dairy based on the DQI-I food group structuring based on calcium content [22] whereas the recommended three servings was set for dairy alone [31]. Each asterisk (*) represents an extreme outlier to the data, for example, two female participants consumed over seven servings of fruit a day and one male participant consumed over eight servings of fruit a day.

**Table 1 nutrients-13-02384-t001:** Equations to determine proportion of low energy reporters (LER).

Age 70+	TEI: BMR ≤ 0.92 = LER	
	Oxford Equation for BMR [36].	% LER
Male (*n* = 190)	13.7BW + 481 (kcal/d)	32.6
Female (*n* = 274)	10.0BW + 577 (kcal/d)	32.5
Total (*n* = 464)		32.5

Abbreviations: total energy intake (TEI), basal metabolic rate (BMR), body weight (BW), kilocalories per day (kcal/d).

**Table 2 nutrients-13-02384-t002:** Differences in demographics and diet quality between plausible reporters and LER.

Demographics and Dietary Variables	Plausible Reporters (*n* = 313)	LER (*n* = 151)	*p*-Value
Age, years	80 (77–84)	80 (76–83)	0.097
Sex, *n* (% female)	185 (59.1)	89 (58.9)	0.973 ^1^
Marital Status, *n* (% partnered)	153 (48.9)	83 (55.0)	0.219 ^1^
Education level, *n*			0.917 ^1^
(% primary)	10 (3.2)	4 (2.6)	
(% secondary)	173 (55.3)	82 (54.3)	
(% tertiary)	130 (41.5)	65 (43.0)	
Housing, *n* (% private)	270 (86.3)	128 (84.8)	0.666 ^1^
Deprivation (NZDep2018)	5 (4–8)	6 (4–8)	0.699
Medical conditions, count	3 (2–4)	3 (2–4)	0.874
Vision, *n* (% impaired)	291 (93.0)	141 (93.4)	0.871 ^1^
Hearing, *n* (% impaired)	291 (93.0)	138 (91.4)	0.546 ^1^
BMI, kg/m^2^	26.9 (24.4–30.1)	30.7 (28.3–34.1)	<0.001
Medications, count	5 (3–7)	5 (3–8)	0.927
Supplements, count	0 (0–1)	0 (0–1)	0.438
Alcohol consumption, *n*			0.788 ^1^
(% Never)	73 (23.3)	37 (24.5)	
(% Occasional)	124 (39.6)	63 (41.7)	
(% Regular)	116 (37.1)	51 (33.8)	
NEADL score	20.0 (18.0–21.0)	20.0 (18.0–21.0)	0.776
DQI-I total score (/100)	60.8 (55.3–65.2) ^a^	58.9 (52.2–63.3)	<0.001 ^2^
Variety score (/20)	14.0 (12.0–16.5)	11.5 (9.5–14.0) ^b^	<0.001
Adequacy score (/40)	30.8 (27.6–33.0)	27.3 (23.3–30.0)	<0.001
Moderation score (/30)	12.0 (9.0–13.5)	13.5 (12.0–16.5)	<0.001
Balance score (/10)	4.5 (3.3–5.5) ^c^	4.8 (3.7–6.0) ^d^	0.054 ^2^
Energy, kcal/kg BW	23.2 (20.2–28.0)	14.3 (11.6–15.6)	<0.001
Protein, g/kg BW	0.9 (0.7–1.1)	0.6 (0.5–0.7) ^e^	<0.001
Empty Calorie, %TEI	37.8 (27.3–48.8) ^f^	30.7 (20.9–39.9) ^g^	<0.001 ^2^
Carbohydrate, %TEI	46.8 (41.9–52.8) ^h^	48.5 (43.0–53.2)	0.075
Protein, %TEI	15.2 (13.4–17.2)	17.5 (14.5–19.9)	<0.001
Fat, %TEI	34.9 (29.4–40.1) ^i^	31.7 (26.0–36.3) ^j^	<0.001 ^2^
Saturated Fat, %TEI	13.6 (11.5–16.4) ^k^	11.8 (9.2–14.7)	0.006

Values presented as median (IQR). *p*-values derived from non-parametric independent samples test unless otherwise indicated: ^1^ Chi-squared tests, ^2^ Independent samples *t*-test. Italicised values reflect higher quality diets relative to comparison group. Normally distributed values mean (SD) as follows: ^a^ DQI-I total score 60.2 (7.8), ^b^ variety score 11.7 (3.4), ^c^ balance score 4.5 (1.5), ^d^ balance score 4.8 (1.5), ^e^ protein 0.6 (0.2) g/kg BW, ^f^ empty calorie 38.1 (14.8) %TEI, ^g^ empty calorie 31.4 (1.5) %TEI, ^h^ carbohydrate 46.9 (8.0) %TEI, ^i^ fat 35.0 (7.4) %TEI, ^j^ fat 32.0 (8.2) %TEI, ^k^ saturated fat 13.8 (3.6) %TEI.

**Table 3 nutrients-13-02384-t003:** Summary of DQI-I scores by sex between all reporters, plausible reporters and LER.

	Male				Female			
DQI-I Scores	Everyone	Plausible Reporters	LER	*p*-Value	Everyone	Plausible Reporters	LER	*p*-Value
DQI-I (/100)	59.27 (53.2–67.9)	60.2 ^a^ (53.9–65.3)	57.3 ^b^ (52.6–62.3)	0.024 ^†^	60.8 (54.3–64.6)	61.0 ^c^ (56.0–65.2)	60.4 (51.8–63.7)	0.008
Variety (/20)	13.5 (11–16.5)	15.0 (12–17)	11.8 ^d^ (9.9–14.5)	0.001	13.0 (11–15)	13.5 (12.0–15.8)	11.5 ^e^ (9.5–13.5)	0.001
Adequacy (/40)	29.4 (26.6–32.3)	30.3 (27.0–33.4)	27.6 ^f^ (23.6–30.4)	0.009	29.8 (26.8–32.1)	30.9 (28.1–32.9)	26.9 (23.2–30.0)	<0.001
Moderation (/30)	12.0 (9.0–15.0)	10.5 (7.5–13.5)	12.0 ^g^ (10.5–16.5)	0.023	12.0 (10.5–15.0)	12.0 (9.0–13.5)	15.0 ^h^ (12.0–18.0)	<0.001
Balance (/10)	4.7 (3.7–6.0)	4.5 ^i^ (3.4–5.7)	5.0 ^j^ (4.1–6.0)	0.020 ^†^	4.7 (3.5–5.5)	4.6 ^k^ (3.3–5.5)	4.7 ^l^ (3.5–6.0)	0.535 ^†^

Values presented as median (IQR). *p*-values compare plausible reporters and LER. ^†^
*p*-value determined by independent samples *t*-test while all other *p*-values derived from non-parametric samples tests. Italicised values reflect higher quality diets relative to comparison group. Normally distributed values mean (SD) as follows: ^a^ DQI-I total score 59.8 (8.7), ^b^ DQI-I total score 56.8 (8.2), ^c^ DQI-I total score 60.5 (7.2), ^d^ variety 11.6 (3.5), ^e^ variety 11.7 (3.3), ^f^ adequacy 27.0 (4.8), ^g^ moderation 13.2 (4.3), ^h^ moderation 14.9 (4.0), ^i^ balance 4.5 (1.5), ^j^ balance 5.1 (1.4), ^k^ balance 4.5 (1.5), ^l^ balance 4.6 (1.6).

## Data Availability

The SUPER Study committee chair, r.teh@auckland.ac.nz.

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
