# Peer review of "The Diet Quality and Nutrition Inadequacy of Pre-Frail Older Adults in New Zealand"

_nutrients, 2021, doi:10.3390/nu13072384_

Round 1
Reviewer 1 Report
Thank you for your interesting study.
This study investigated the diet quality of pre-frail community-dwelling older adults in New Zealand. As a result, their diet quality was moderately high in variety and adequacy but poor in moderation and balance. I think this study is interesting and important for future studies and policymaking to prevent frailty in communities. I have some minor comments below.
Minor comments
- The title should be added “New Zealand”. It would be helpful for readers.
- Line 96: You should add the reason that the definition of older people is different by race. Many readers outside your country may wonder about it.
- Line 246: Regarding Multivariate analyses, you should show the models constructed. What variables were adjusted?
Author Response
Response to Reviewer 1 Comments
for Nutrients-1280775
Thank you for your review and the useful comments which have improved the manuscript.
Please see our responses below:
Please note, line numbers were recorded when the “All Markup” view was enabled.
Comment 1: The title should be added “New Zealand”. It would be helpful for readers.
Response: "in New Zealand" has been added to the title (line 3).
Comment 2: Line 96: You should add the reason that the definition of older people is different by race. Many readers outside your country may wonder about it.
Response: We have added the reason for the different age threshold for Maori and Pacific people in line 97 -99 "A lower age group was chosen for Māori and Pacific people as there is disparity in health status and life expectancy compared to the general population and levels of disability occur at a younger age for these groups [27]."
Comment 3: Line 246: Regarding Multivariate analyses, you should show the models constructed. What variables were adjusted?
Response: Tables of results from multivariate analyses have been added to Supplementary files (Table S3).
“Regression models conducted using multiple forward linear regression. All models were adjusted for age, sex, marital status, education, living arrangement, deprivation, medical conditions, vision, hearing, medications, supplements, alcohol, NEADL, and total DQI-I score except for total DQI-I score in model 2.”
Reviewer 2 Report
This is a novel observational study of nutrition tendency in pre-frail older adults. The study design, statistical analysis and these scientific set up appears very solid. I enjoyed reading this manuscript but one question remains unanswered. How is this deemed to be generalizable to the rest of the world? Not that I imagine that diet in New Zealand is different from other developed countries, but it requires knowledge about diet in New Zealand. Are there any particular food or customs enjoyed in New Zealand? I heard that molasses consumption also high (as well as in Australia)? Do older people love McDonald's as in the US? These factors may be ameliorating or worsening the imbalance. Readers in the rest of the world need help to translate/localize this finding in their own part of the world.
The authors missed to fix "(Error! Reference source not found.)" I know this tends to happen when we apply MDPI's format. I hope this can be fixed in the next round.
Author Response
Response to Reviewer 2 Comments
for Nutrients-1280775
Thank you for your review and the useful comments which have improved the manuscript.
Please see our responses below:
Please note, line numbers were recorded when the “All Markup” view was enabled.
Comment 1: How is this deemed to be generalizable to the rest of the world? Not that I imagine that diet in New Zealand is different from other developed countries, but it requires knowledge about diet in New Zealand. Are there any particular food or customs enjoyed in New Zealand? I heard that molasses consumption also high (as well as in Australia)? Do older people love McDonald's as in the US? These factors may be ameliorating or worsening the imbalance. Readers in the rest of the world need help to translate/localize this finding in their own part of the world.
Response:
Thanks for your question.
Foods or customs in New Zealand tend to be similar to other western countries reflected in the common treats like sweets, biscuits, baked goods. New Zealanders tend to also like pies, sausages and dairy given the local food industry. Eating out is more common for the younger generations rather than our sample of older adults who generally prefer home-cooked meals. Molasses are growing in popularity as a “health food” for younger generations and is not widely seen in our sample (roughly three person consumed molasses at baseline). To acknowledge this point, the following sentences were included in line 393-404.
“The index is sensitive to identify undernutrition and overnutrition, which is seen in both developed countries like New Zealand and developing countries [22]. In this regard, our results allow for future international comparisons. We still advise caution in generalising the findings to other populations which may have different dietary cultures and environments. Most of our study sample are of British descent and still follow closely to the traditional homemade ‘western diet’. As seen in the latest New Zealand nutrition survey (2008/9), less than 0.5% of adults aged over 71+ have fast food or takeaways for three or more times a week [31]. It is therefore more likely that the dietary imbalance in New Zealand arises from a high consumption of ‘home foods’ like biscuits, cakes, desserts, sweet drinks, sweets, butter and margarine, sweet spreads, meat or cheese pies and pastries, sauces, processed meats and alcoholic drinks [31,32].
Comment 2: The authors missed to fix "(Error! Reference source not found.)" I know this tends to happen when we apply MDPI's format. I hope this can be fixed in the next round.
Response: We have now amended these references as they referred to Supplementary tables and figures.